# Passively Sampled Ambient Hydrocarbon Abundances in a Texas Oil Patch

**Olivia M. Sablan [1], Gunnar W. Schade [2,*] and Joel Holliman [2]**

[1]   Department of Chemistry, Carroll College; 1601 N Benton Ave., Helena, MT 59625, USA; osablan@carroll.edu
[2]   Department of Atmospheric Sciences, Texas A&M University, College Station, TX 77843, USA; joel52@tamu.edu
*    Correspondence: gws@geos.tamu.edu

**Abstract:** The United States has experienced exceptional growth in oil production via unconventional extraction for over a decade. This boom has led to an increase in hydrocarbon emissions to the atmosphere. With Texas as the leading contributor to growing oil production, it is important to assess the effects the boom has had on the environment and communities at local and regional levels. We conducted a pilot study to investigate the use of passive samplers for evaluating potential off-site risk from hydrocarbon emissions in a relatively low production activity area of the Texas Eagle Ford shale. Emissions from production sites include benzene, a hazardous air pollutant and known carcinogen. Passive hydrocarbon sampling devices (Radiello samplers) were used to monitor hydrocarbon levels on a rural property near a production site with an occasional flare for one year. Selected hydrocarbons were analyzed using thermal desorption and gas chromatography with flame ionization detection. Benzene concentrations were found to be correlated with changes in season, with higher abundance in the winter months. Benzene levels at this site were similar or higher than those observed in urban areas, away from shale oil and gas production. Increased benzene concentrations were distinguished when winds advected hydrocarbons from the production site, suggesting that oil and gas site emissions have a greater impact on the local community when winds advect them towards those living downwind; however, hydrocarbon levels in this low production area never exceeded state air monitoring comparison standards.

**Keywords:** passive sampling; hydrocarbons; exposure; shale oil production; air pollution

## 1. Introduction

U.S. oil production has recently exhibited rapid growth due to unconventional extraction methods [1], using hydraulic fracturing of horizontally drilled well bores [2], a technological combination commonly identified as "fracking". Production is expected to continually set annual growth records through 2027 [3]. With this trend in oil production, concerns for the effect this development has on the environment and the communities nearby arise. The state of Texas, as a top producer of unconventionally extracted oil from shale resources, sits at the center of these environmental and social impacts.

The rapid growth of unconventional oil and gas (UOG) production in the Texas Eagle Ford shale (https://www.rrc.state.tx.us/media/55436/eagle-ford-oil.pdf) has led to an increase in fugitive hydrocarbon emissions [4,5]. The main sources of these hydrocarbon releases at oil and gas production sites are from liquids storage tanks and equipment leaks [6–10]. In addition, raw natural gas, when produced as a byproduct of unconventional oil production, is often burned off at the well site in a flare. Inefficient flares emit substantial amounts of raw gas, but even efficiently burning flares (98 + % conversion to $CO_2$ [11]) may produce a considerable amount of additional air pollution in the form of unsaturated hydrocarbons and partially oxidized volatile organic compounds [4,12,13], as well

as nitrogen oxides and black carbon [14–16]. Aside from direct environmental impacts at the site of emission, the atmospheric oxidation of hydrocarbons produces ozone, a secondary air pollutant with significant environmental and public health effects [17,18].

Volatile non-methane hydrocarbons (NMHCs) released from oil and gas production sites include several hazardous air pollutants (HAPs). A common HAP of national importance is benzene, a carcinogen [19–21]. HAPs, such as benzene, are a potential danger to public health in shale production areas [22–25]. To evaluate this risk, it is important to determine the exposure to NMHCs, especially HAPs, in areas where dense oil and gas production may significantly increase population exposure to fugitive emissions. Due to its toxicity, benzene was selected as a target compound in this study. Other hydrocarbons and HAPs of interest included n-hexane and toluene.

Since the measurement of atmospheric NMHCs most often involves in situ sampling followed by laboratory-based analysis, most previous work documented in the literature describes one of two assessment methodologies: (i) high frequency (typically 1-h resolution), most often stationary sampling and in situ measurement at an air quality monitoring station (AQM); or (ii) low frequency (e.g., weekly) ambient air collection into canisters ("grab sampling") with subsequent, detailed laboratory-based analysis. Both methods can be energy- and labor-intensive, especially if carried out at multiple locations. However, the vast spatial extent of shale production areas in the U.S. mean these traditional methods of NMHC analysis are not only costly, but also mostly inadequate to determine representative exposure situations, because neither the emission sources or the emissions themselves, are randomly distributed, nor are the people living in these UOG production areas. Data from an AQM may represent exposures of people living nearby, but not of people living farther away, or of people living closer to the emitting production sites. On the other hand, grab sampling, which can be done anywhere, may represent exposure at the time of sampling but may not represent exposure over longer periods of time. We, therefore, decided to test passive air monitoring [26] as an economically more efficient way to study NMHC exposure.

For this pilot study, concentrations of hydrocarbons were measured weekly using passive air sampling at a private property in close proximity to an unconventional oil and gas production site. These measurements were used alongside regional weather data to establish dispersion patterns. The area of study was in the Eagle Ford shale of Texas, and samples were taken from September 2018 to September 2019. Our main objectives were to test the feasibility of a simple deployment and sampler turn-around strategy with the help of a volunteer, and to note challenges and strengths while determining selected, average HAP exposure levels in an oil and gas production environment for comparison with standard air quality monitoring.

## 2. Methods

### 2.1. Location and Sampling

Passive air sampling with Radiello samplers [27,28] was used to monitor off-site hydrocarbon concentrations. The sampling process is similar to the EPA's (United States Environmental Protection Agency) Method 325A for monitoring VOCs (Volatile Organic Compounds) [29]. The passive NMHC sampling method has been shown to be reliable for determining selected hydrocarbon concentrations by the EPA and others [30–33], and has previously been used in shale areas [34].

Radiello samplers containing Carbograph 4 sorbents were placed inside yellow diffusive bodies (all Radiello materials were purchased from Sigma-Aldrich, USA), underneath a protective shelter to keep out rainwater, approximately one meter above the ground (Figure 1). Two diffusive bodies (replicates) were placed under the shelter on a volunteer's private property approximately 100 meters from the nearest equipment on an oil well site. A third sampler was kept onsite inside its protective glass container as a field blank for quality assurance purposes. Three sets of samplers were used: one in deployment, one in reserve (spare), and one undergoing analysis. The volunteer was trained to exchange the cartridges once weekly and replace them with the spare set, then bring the exposed set to our laboratory at Texas A&M University in College Station, Texas, in exchange for the next spare

set. Analyzed and pre-cleaned cartridges (heating for 20 min at 220 °C under high purity $H_2$ flow) were stored for up to a week in their glass containers, then deployed for one week during a 52-week span from September 2018 to September 2019. To avoid bias and assess the quality of the sorbents, the cartridges and their diffusive bodies were randomly selected each week.

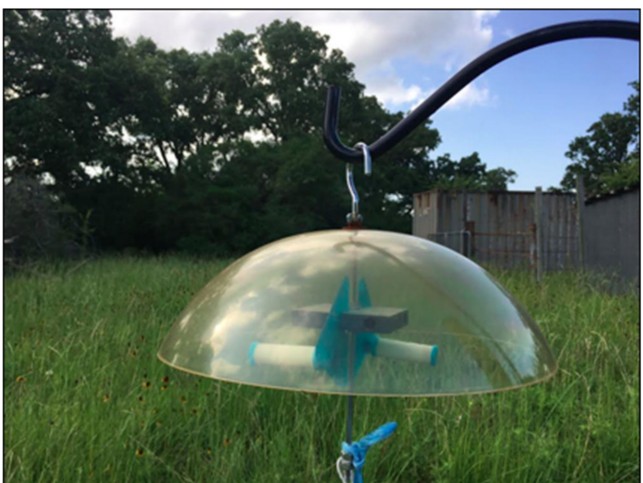

**Figure 1.** Sampling setup with Radiello diffusive bodies under the protective shelter.

The studied oil and gas production site, located toward the northeast end of the Eagle Ford shale in south central Texas, was northwest of the samplers (Figure 2). The site was observed to have an occasional flare in its northwest corner, which was located approximately 200 meters from the sampling location. The small flare, along with venting storage tanks and potentially leaky pipes and other equipment, was suspected to provide the majority of local exposure to hydrocarbons, because car traffic density as the commonly major source is very low in this rural area (~200 cars per day). The nearest main road, State Highway 30, lies 5.5–6 km to the site's south and southwest, and bears only 7500–9500 annual average daily vehicle traffic (Texas Department of Transportation (TxDOT), https://www.txdot.gov/inside-txdot/division/transportation-planning/maps.html). The nearest urban area, Bryan (College Station), with approximately 200,000 residents, lies 20 km to the west, a wind direction very rarely encountered in east Texas.

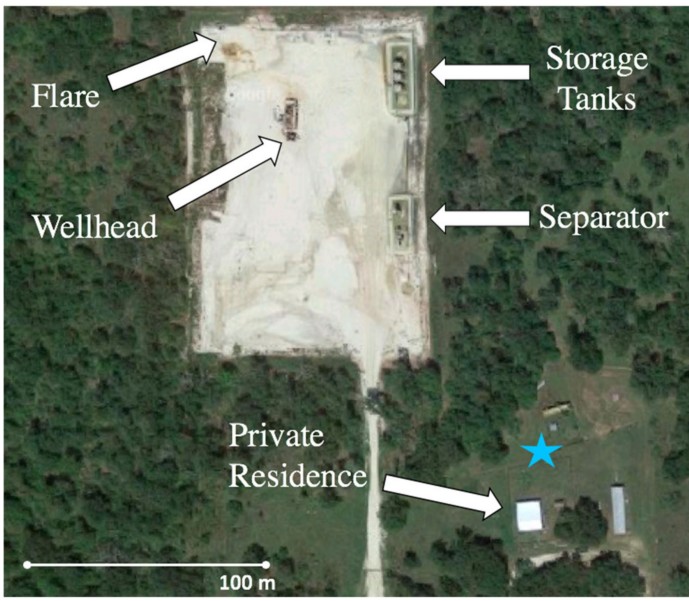

**Figure 2.** Bird's eye view of the sampling location (indicated by the blue star).

## 2.2. Sample Analysis

Samples were analyzed using thermal desorption and gas chromatography with flame ionization detection (TD-GC-FID). The Radiello cartridges were transferred into standard $1/4$ inch OD metal tubes and processed using a Perkin-Elmer ATD400 autosampler [35]. Calibration samples, produced weekly by actively sampling a diluted standard mixture containing toluene and hexane (Scott-Marrin Inc., Riverside, CA, USA) onto standard glass cartridges filled with similar adsorbents, were processed alongside the field samples. Samples were desorbed at 200 °C for ten minutes under high purity hydrogen flow from a generator (Matheson TriGas, Montgomeryville, PA, USA), pre-concentrated onto a cooled microtrap filled with Carbotrap X inside the ATD400, then rapidly desorbed on-column. A 60 m, 0.25 mm ID Rtx-624 column was used for compound separation with high purity hydrogen as carrier gas. Analytical precision using this method for most hydrocarbons at the ppb level was 2% or better, while accuracy was 5% or better based on the standard's accuracy. Quality control included three types of blanks: the field blank, an unexposed Radiello cartridge; a laboratory blank, an adsorbent cartridge (glass) stored in the laboratory; and a system blank, an empty glass cartridge.

In our experience, the main uncertainties in Radiello passive sampling stem from the variability and accuracy of the accumulation rate of hydrocarbons onto the selected adsorbent. Radiello samplers have a large accumulation rate, and thus higher sensitivity than samplers used in EPA Method 325. However, the adsorbent may saturate under higher exposure or the stated accumulation rate (given in mL per minute; e.g. 27.8 mL min$^{-1}$ for benzene) by the manufacturer for a given hydrocarbon may be biased under the applied exposure conditions. Based on our own laboratory and field testing, we estimate that the accuracy of passively sampled hydrocarbon concentrations using Radiello samplers is of the order of 25% or better. Laboratory studies are ongoing to validate manufacturer-listed accumulation rates at near-ambient levels and to determine rates for hydrocarbons that are not listed.

The hydrocarbons reported here include 2-methylpentane, 3-methylpentane, n-hexane, benzene, and toluene. Manufacturer-listed accumulation rates were used for the latter three compounds, while rates identical to n-hexane were used for the branched hexanes. Ambient levels are reported in parts per billion, converted from the micrograms on each cartridge using standard temperature and pressure conditions. A temperature correction to the sampling rate was made according to the Radiello manual using the weekly average air temperature for each sample week.

## 2.3. Auxiliary Data

Weather data was collected from a Mesonet station located 30 km to the west-southwest of the site, operated by Texas A&M University. New observations were recorded by the station hourly, and we used the hourly average temperature and wind data from the station as representatives of temperature and flow conditions at the hydrocarbon collection site. Based upon comparisons to neighboring Mesonet sites (not shown), our weekly assignments of hourly wind directions to one of four quadrants was generally reproducible to within 10% of the absolute value, despite tens of kilometers distance.

## 3. Results and Discussion

We report all hydrocarbon levels as the average between the two replicates, unless only one replicate was available. No significant contaminations were observed in the system or laboratory blanks. However, benzene did show a variable amount in the field blank. The average field blank level in our samples represented 30% of the average ambient levels. Since the blank peak was inconsistent in size and among Radiello samplers, it remains unclear whether all samplers could have been affected similarly. We interpret this variable contribution as unknown compounds underlying the benzene peak, possibly from contamination of an as yet unknown origin entering the sample during storage and transport. Therefore, no correction was made to the data.

Statistics of the evaluated hydrocarbons for the 52-week sampling period are listed in Table 1. These concentrations should be compared to the air monitoring comparison values (AMCV) provided

by the Texas Commission on Environmental Quality (TCEQ) (https://www.tceq.texas.gov/cgi-bin/compliance/monops/agc_amcvs.pl). The TCEQ states that hydrocarbon concentrations surpassing the values in Table 1 have the potential to be harmful to human health under long-term exposure. None of the levels of hydrocarbons of interest from this study exceeded the Texas AMCV values.

**Table 1.** The volume mixing ratio statistics of hydrocarbons measured in this study compared to air monitoring comparison values (AMCV) provided by the Texas Commission on Environmental Quality (TCEQ) [36].

| Compound [1] | Average (ppb) | Median (ppb) | Maximum (ppb) | Long Term AMCV (ppb) | Average CoV Of Duplicate Samples |
|---|---|---|---|---|---|
| 2-methylpentane | 0.32 | 0.28 | 1.29 | 190 | 0.12 |
| 3-methylpentane | 0.17 | 0.15 | 0.67 | 190 | 0.12 |
| n-hexane | 0.30 | 0.28 | 0.99 | 190 | 0.14 |
| Benzene | 0.31 | 0.25 | 1.32 | 1.4 | 0.21 |
| Toluene | 0.07 | 0.06 | 0.23 | 1100 | 0.32 |

[1] Minimum levels in all cases were close to the detection limits of 10-20 ppt per compound.

All hydrocarbon levels observed were comparatively moderate to low. Particularly, benzene levels generally remained below 1 ppb, as shown in Figure 3. Included in this figure are several samples throughout the year that were affected by high amounts of humidity, many of which were due to rainfall. This is despite the adsorbent's low affinity to water vapor, its equilibrium-based collection onto the cartridge, and the subsequent processing of substantial amounts of water vapor through the chromatograph causing delayed retention times and flame ionization detection (FID) flameouts during sample analysis, which either caused larger uncertainties due to lower peak resolution and replicate reproducibility or data loss. Despite manual re-ignition, we tracked the "wet" samples, highlighted in Figure 3. As no clear distinction in hydrocarbon concentrations was found between the "wet" samples and the unaffected samples, the "wet" samples were included in the remainder of the results but were distinguished for transparency (Figure 3). These instances, nevertheless, highlight a major challenge of this passive sampling method in humid environments.

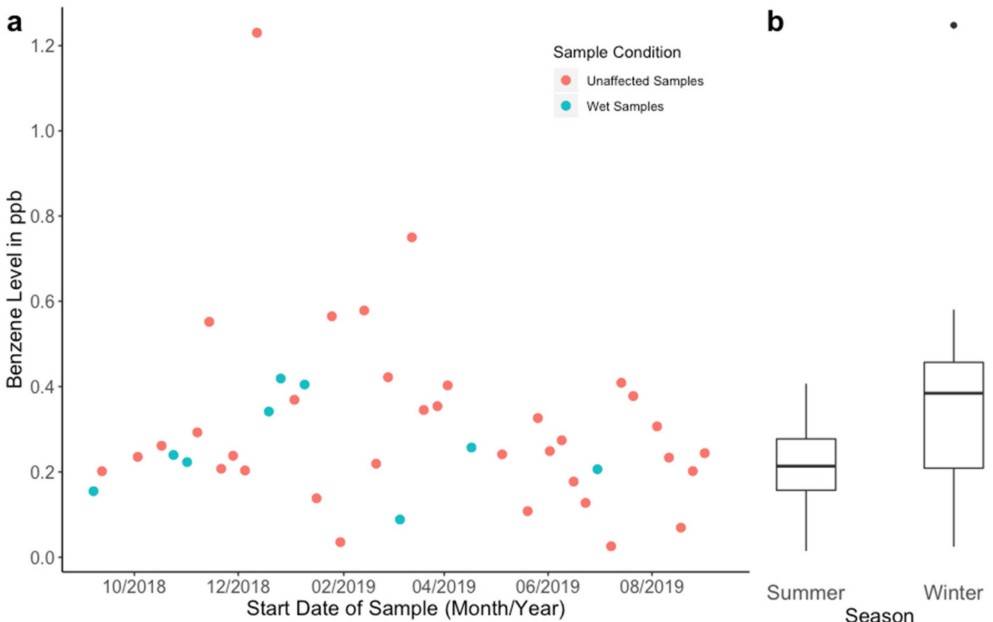

**Figure 3.** Seasonal changes in benzene levels. Each point (**a**) is plotted by the start date of the sampling week. "Summer" and "winter" value boxplots (**b**) encompass three months of data, and were statistically significantly different at the 90% confidence level ($p < 0.07$).

Relationships between reported hydrocarbons over the 52-week campaign are shown in Figures 4 and 5. Strong correlations can be seen between the investigated alkanes (Figure 4). This is likely due to these alkanes originating from a common emission source, such as the evaporation from gasoline tanks at nearby oil and gas production sites. Weaker correlations were found between the alkanes and the aromatics (Figure 5), likely because their sources are more variable. As an example, they both can occur as evaporative emissions from storage tanks and in car exhausts, while toluene is also a common solvent. Additional causes of higher variability (higher CoV for the aromatic compounds, Table 1) could come from the more variable accumulation rates of aromatics compared to alkanes. We note, however, that toluene-to-benzene ratios were generally below one. This contrasts with urban areas, which normally display ratios of around 2 or higher [37] based on car traffic as the dominant local emission source. This comports with an additional benzene source in the Eagle Ford shale area, as discussed by Schade and Roest [38].

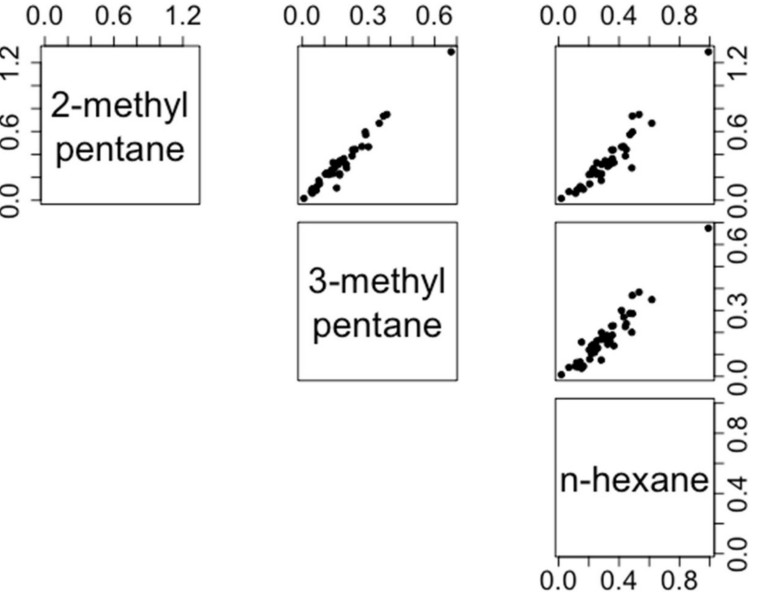

**Figure 4.** Relationships between C6-alkanes (in ppb) during the 52-week sampling period. All Spearman's rank correlations exceeded 0.9.

Here, we briefly discuss benzene concentrations because of its toxicity. Benzene levels in ambient air have been dropping for decades, because its dominant emission source, car traffic, has been more strictly regulated over the years, such as through reductions in the benzene content of gasoline [39]. Thus, ambient benzene levels in urban areas, where sources are still dominated by dense car traffic, have reached their lowest levels in decades. In Table 2, we compare benzene and hexane levels from three Texas AQM stations to our passive sampler data.

**Table 2.** Average annual volume mixing ratios of benzene and n-hexane measured at urban air quality monitoring (AQM) stations in Texas compared to rural results in this study.

| AQM Station | Benzene (ppb) | | n-hexane (ppb) | | Site ID (AQS) |
|---|---|---|---|---|---|
| | 2017 | 2018 | 2017 | 2018 | |
| Dallas Hinton [1] | 0.12 | 0.11 | 0.15 | 0.16 | 481130069 |
| Houston Bayland Park [2] | 0.22 | 0.18 | 0.14 | 0.15 | 482010055 |
| Houston Haden Road [1,3] | 0.36 | 0.39 | 0.51 | 0.58 | 482010803 |
| This work (2018/2019) | 0.31 | | 0.30 | | NA |

[1] In situ hourly measurements from an auto-GC. [2] The 24-hr canister samples collected every six days. [3] The Haden Road site in the Houston ship channel is bordered to the south by the petro-chemical industry, and to the north by a residential neighborhood (Cloverleaf).

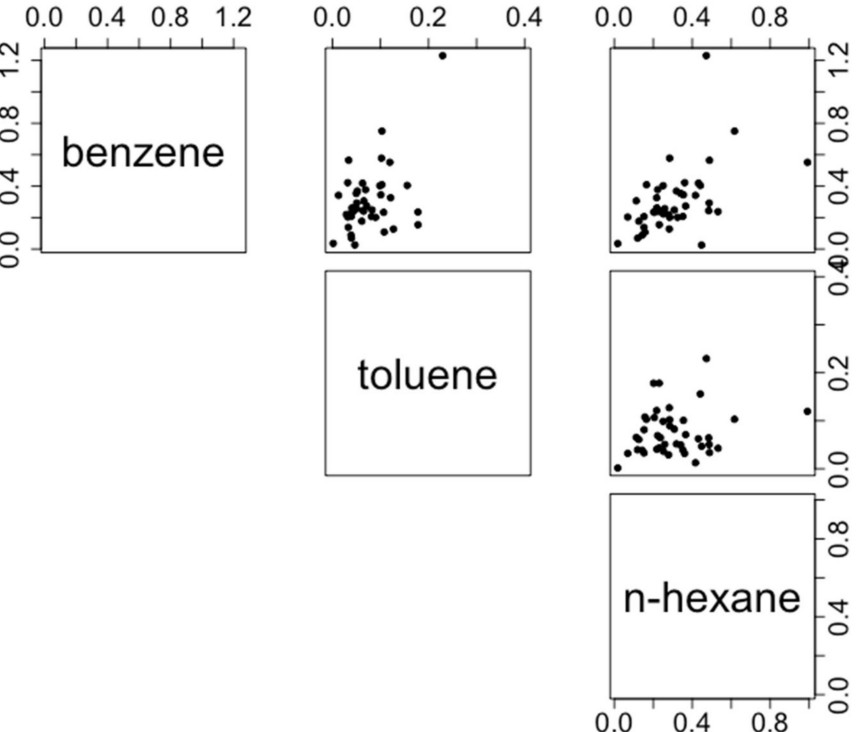

**Figure 5.** Relationships between aromatics and non-aromatics (in ppb). In this case, Spearman's rank correlation coefficients ranged from 0.27 (toluene-benzene) to 0.56 (benzene-hexane).

For instance, the annual average benzene abundances at the heavily trafficked, central Dallas Hinton AQM (using in situ hourly measurements) were 0.12 and 0.11 ppb in 2017 and 2018, respectively. Nearly twice these levels were found in southwest Houston (Bayland Park site) using lower frequency 24-h canister sampling. In comparison, average benzene levels at the Haden Road site in Houston, which is impacted by petro-chemical industry emissions, and in our study at a rural site with essentially no traffic were higher. This could be explained by the renewed and rapidly increasing production of oil and gas in shale areas, which has led to a new, evaporative source of benzene that can dominate the traffic source [38,40].

To evaluate whether we can observe the nearby hydrocarbon source at our sampling site, we investigated seasonal changes and advection. During the winter months, we observed an increase in benzene concentrations (Figure 3). This finding is likely the result of seasonal changes, including larger fractions of air transport toward the sampler from the well site, as well as generally lower boundary layer depths at those times of the year. In east Texas, an average wind shift occurs during the winter months from a dominance of southerly winds to a dominance of northerly winds, largely as a result of a southward move of the polar front. During the warmer summer months, hydrocarbon concentrations are generally lower because hydrocarbon reaction rates with OH radicals in the atmosphere are higher and deeper boundary layers cause stronger dilution.

When benzene levels were compared to the percentage of weekly winds from the northwest sector, including calm conditions, we found a significant dependence, as shown in Figure 6. Northwesterly winds bring emissions from the oil production site to the sampler location (Figure 2), and calm conditions were included to account for the drifting of hydrocarbon emissions toward the samplers when the wind fails to disperse them effectively. The significant increase with higher percentages of northwest and calm winds suggests that hydrocarbon emissions from the site have a greater impact on off-site exposure when winds from the site flow towards those living downwind or when the conditions are calm, allowing local accumulation of emissions. This correlation, though present in most seasons of the year (Figure 6), was highly variable, likely because (i) the fraction of

NW winds in this area may have significantly differed from that measured 30 km away; and (ii) the well-site was not a regular and high producing location, and thus likely not a steady high emissions source. In addition, dispersion, which is affected by wind speed, is not accounted for in this analysis.

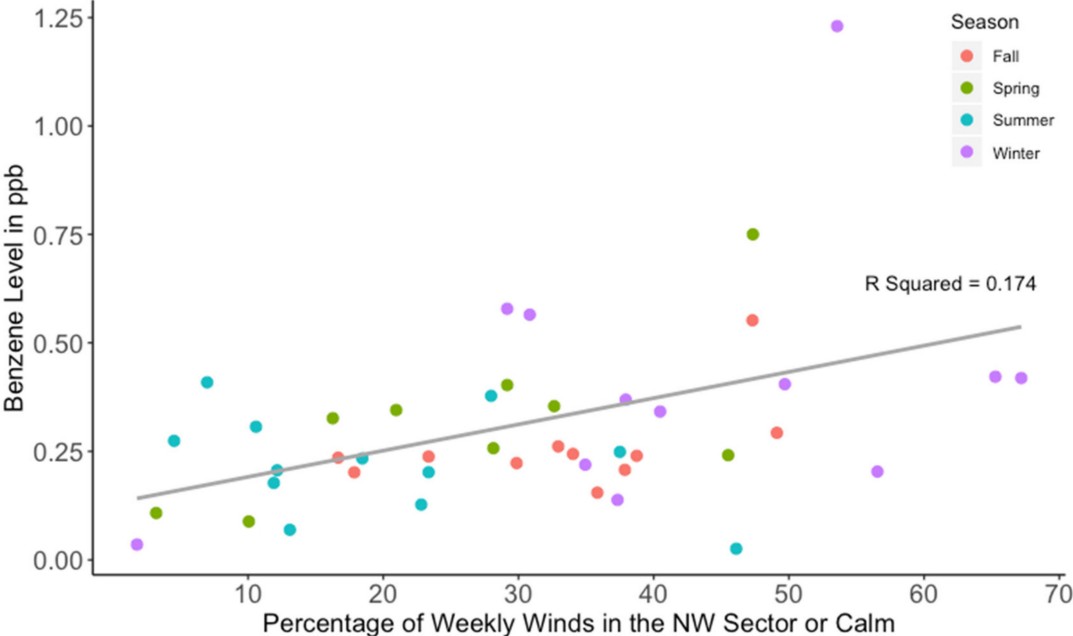

**Figure 6.** Relationship between the percentage of hourly winds traveling over the site toward the monitoring station and benzene levels. As the amount of transport from the site increased, levels of benzene became higher ($p$ = 0.005). Colors correspond to the 3-month season the sample was taken in.

To explore the effect of winds on the dispersion of emissions, wind distribution stick plots were created for each week showing wind vectors for every hour of the sampling week. For instance, during the sampling week starting 27th February, 2019, 61% of winds came from the northwest sector and 4% were calm (Figure 7a). For that week, the average level of benzene was 0.47 ppb. During the week of 16–23 June 2019, a period of typical summer southeasterly flows, only 5% of winds came from the northwest and 7% of winds were calm (Figure 7b). The average level of benzene in that case was only 0.16 ppb. We note that NW winds typically have higher wind speeds (compare Figure 7a with Figure 7b), thus disperse pollutants more effectively than southeasterlies, and therefore one would not expect the hydrocarbon exposure to increase proportionally with the increase in northwesterly winds under the given conditions. Lastly, we observed no cases (weeks) during which winds blew consistently from the NW sector, but we can surmise from Figure 6 that the average benzene exposure under sole NW transport from the nearby emissions site could be 0.2–1.1 ppb (based on our regression prediction model).

In addition to wind direction, benzene levels were also significantly correlated with temperature, as shown in Figure 8. This may be interpreted as the seasonal effect of lower boundary depths during the cooler season, for which temperature may serve as a proxy.

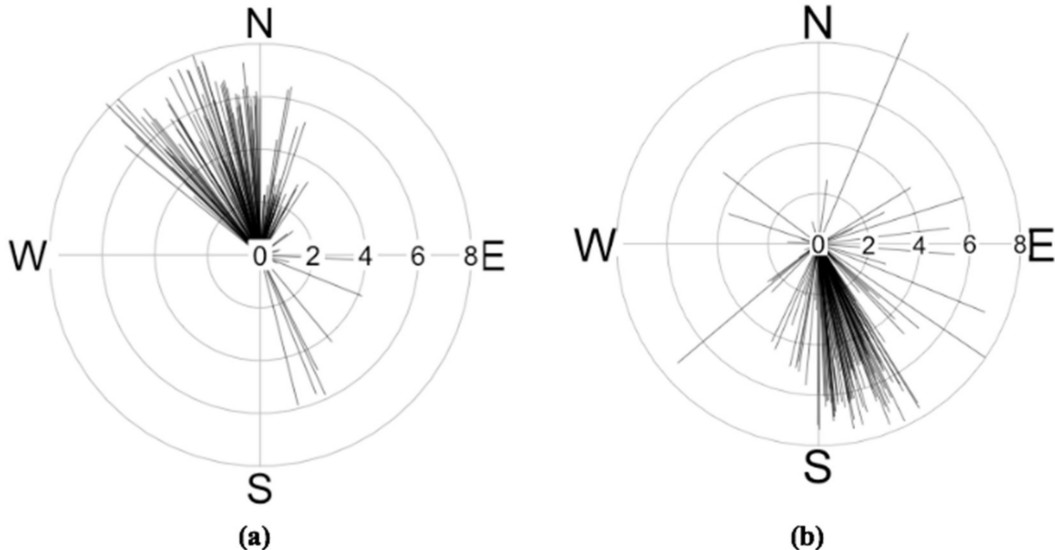

**Figure 7.** Wind distribution stick plots. Each "stick" represents the average wind direction and speed (in meters per second) during one hour of the indicated week (168 hours); 65% of all sticks fell in the NW sector (**a**), while only 5% fell in the NW sector (**b**).

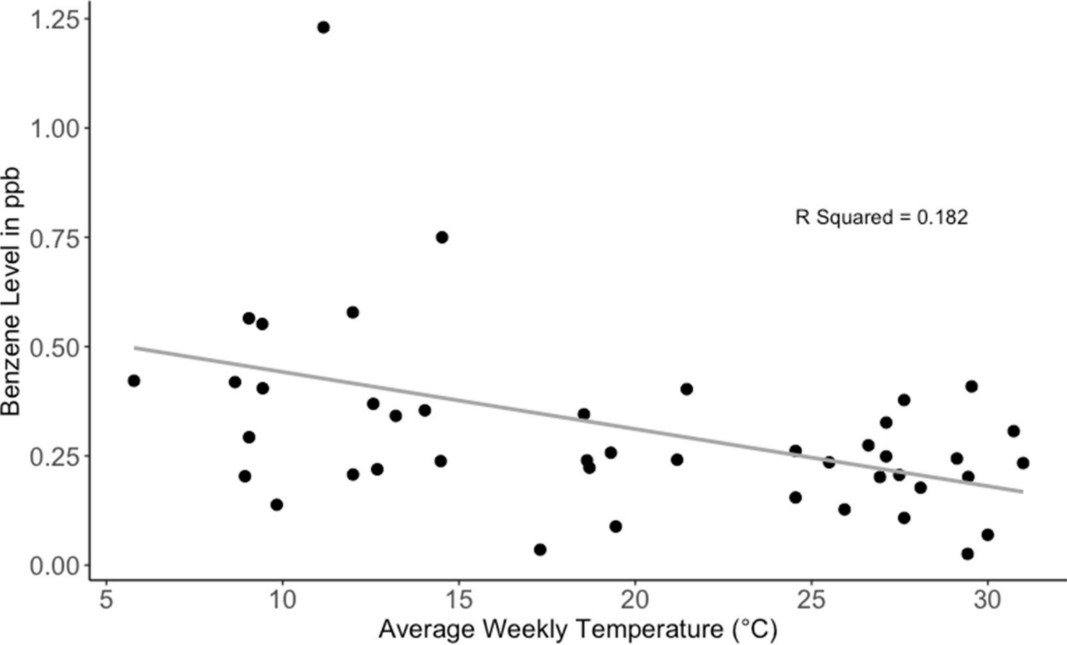

**Figure 8.** Relationship between 2–m air temperature (at a 30–km distant weather station) and benzene levels. The significant negative correlation ($p = 0.004$) is likely due to seasonal boundary layer effects, which co-varies with wind direction.

## 4. Conclusions

A pilot project of one year of passive air sampling of selected hydrocarbons in the Eagle Ford Shale area was conducted on a private property near an oil and gas production site to assess the land owner's outdoor exposure. We encountered several difficulties in this study related to moisture accumulation on the samplers and field blank effects, which prompted us to make modifications to ongoing studies, such as a change in the chromatographic column and the initiation of passive sampling next to a state air quality monitoring station with in situ, hourly measurements, so as to assess passive sampling

measurement accuracy and potential biases as compared to rigorously quality-assured data from the state of Texas.

In general, however, our study—supported by a citizen science volunteer who changed (and delivered) samplers on a weekly basis—shows evidence of passive sampling providing dependable results with comparatively low costs and maintenance. The results presented primarily suggest a seasonal relationship of higher benzene levels during winter months, when this east Texas region experiences a shift in wind patterns from virtually solely southerly winds in the summer to more frequent northerly winds. Lower boundary layer depths during the cooler winter months alongside slower chemical removal cause higher near-surface abundances of hydrocarbons. Levels of benzene, which was targeted due to its high carcinogenicity, were below a statewide long-term air monitoring comparison value based on benzene's 1:100,000 lifetime cancer risk [36]. However, when analyzed with weather data to investigate abundance patterns, higher benzene levels at times the exposure site was downwind from the expected emission source suggested higher exposure levels as a result of local emissions, similar to previous work [34].

Although the hydrocarbons analyzed did not exceed TCEQ's standard for health effects from long-term exposure, our study was carried out in an area of the Eagle Ford shale with comparatively low well pad density. Furthermore, the neighboring well site had comparatively low production volumes in 2018/19, only a small sized flare, and very little traffic associated with oil storage tank draining. Nevertheless, average benzene abundances of 0.3 ppb (0.2 ppb under conservative blank assumptions) were comparable to or higher than average annual levels observed in Texas metropolitan areas that are not affected by nearby oil and gas production, but which carry much higher vehicle traffic. Such levels and seasonal changes are similar to results from the rural Karnes City AQM southeast of San Antonio, which is located closer to the Eagle Ford shale [38], but not near active production sites.

To further evaluate how representative and practical passive sampling of hydrocarbons is compared to the current methodologies, we have initiated two additional volunteer projects serving up to six additional sites, including one near a state AQM. The costs of these projects are substantially lowered through volunteer citizen scientist involvement, and are driven largely by sampler acquisition, replacement, and shipping, presuming labor and analysis costs will be similar to existing methodologies. The results of these projects may provide a better overview of regional differences in exposure as affected by nearby oil and gas emission sources, and when compared to more maintenance and higher cost, as well as possibly non-representative AQM stations. Presuming all of Karnes County (~2000 km$^2$) in the center of the Eagle Ford was to be targeted for extensive monitoring, where a single site would be representative of a ten square kilometer area, six Radiello samplers ($300) would be used for each site with a 10-day turn-around, and where sample shipments would occur in both directions at $20 per shipment but combined over four sites through volunteer efforts, such a project could be accomplished for approximately $150,000, plus minor costs for site shelters and sample packaging. However, the organization of such an effort would be more challenging, requiring 50 volunteers plus the processing of 600 samples per week, since the typical sample processing time is 45-60 minutes. Therefore, larger scale monitoring efforts would likely be limited by field sample processing and associated quality control efforts.

**Author Contributions:** Data curation, O.M.S. and J.H.; Resources, G.W.S.; Supervision, G.W.S.; Visualization, O.M.S.; Writing—original draft, O.M.S.; Writing—review & editing, G.W.S. All authors have read and agreed to the published version of the manuscript.

**Funding:** This research was funded by National Science Foundation: AGS-1559895.

**Acknowledgments:** We are indebted to Debbie Patranella, our citizen scientist volunteer. We particularly thank her for exchanging and transporting samplers on a regular basis, without which this project would not have been possible. Thanks also to Kimberly Sayprasith and Lily Wu for their help analyzing samples and supporting the progress of this project. In addition, we thank the National Science Foundation for funding O. Sablan during a summer Research Experience for Undergraduates (REU) at Texas A&M University, grant AGS-1559895.

**Conflicts of Interest:** The authors declare no conflict of interest.

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
