# Peer review of "Passively Sampled Ambient Hydrocarbon Abundances in a Texas Oil Patch"

_atmosphere, doi:10.3390/atmos11030241_

Round 1

Reviewer 1 Report

The revisions now suffice.  Limitations of the study are now emphasized in the conclusions.

Reviewer 2 Report

The authors were responsive to my comments and I recommend the revised manuscript for publication.

This manuscript is a resubmission of an earlier submission. The following is a list of the peer review reports and author responses from that submission.

Round 1

Reviewer 1 Report

The paper “Passively Sampled Ambient Hydrocarbon Abundances in a Texas Oil Patch” by Sablan et al. presents select VOC concentrations collected with Radiello passive samplers at a site in the vicinity of a small oil-and-gas well pad for one year.  The paper has major limitations relative to application of the passive samplers and interpretation of the data.  It is this reviewer’s opinion that a method expert familiar with Radiello passive samplers should provide comment on the limitations and VOC concentrations reported.

In addition, one monitoring site at a distance of 100 m or more from the well pad operations with many trees as a barrier likely means other influences (perhaps from the volunteer’s property itself) also played a role in the VOCs collected.  For example, the volunteer’s car(s) on the property may have been a VOC influence.  Below are specific concerns:

Lines 66-69: As stated above, one site well away from the well pad’s fenceline and blocked by trees will lower VOC impact.  As such, other influences may have been a factor.  Ideally, an additional site at the well pad’s fenceline perimeter should have been conducted.  If permission to do fenceline monitoring was not provided by the operator, then an additional site further away in a neighboring property should have been established to verify any decline in VOCs and overall background levels.  This needs to be mentioned as a limitation.

Lines 94-104: Please provide the UTM coordinates for this site.  The description of the site location is very difficult to pinpoint from what is provided.

Lines 118-128: In terms of accuracy, were Radiellos compared with reference canister samples?  Was the accuracy based on spiked samples?  These points are important since the Radiello is a research-grade device.  Additional detail needs to be provided regarding how sampler accuracy was assessed.

Lines 136-139: Meteorology was collected 30 km (almost 19 miles) away from the well pad site, not at the site or at a closer meteorological station.  This is a limitation when evaluating pollutant levels by wind direction since the meteorological station used was too far to assess local winds.  Also, winds will likely have come from all directions (including meandering winds) when compared with the Radiello week-long sampling integrals.  My suggestion is that you remove the presentation of VOCs by wind direction.

Lines 141-145: It appears that the authors have not sufficiently conditioned their tubes since they state that they have a variable blank background of benzene (average of 30%).  They need to discuss that they could have done better on the blank levels by conditioning their tubes more stringently.  I am not an analytical methods expert but if 30% of the benzene concentrations are due to sampler (not source) issues, this would be a major interference in the reported VOC concentrations.  This would also affect method minimum detection limits (MDLs).

Table 1: MDLs should be provided.  Minimum and median values measured should be reported in separate columns.  On this table or another table, duplicate precision should be provided for each VOC.  If coefficients of variation from the duplicates are high, then the reported variabilities due to temperature, season, etc. may have been affected by method precision.

Table 1: For toluene-to-benzene ratios, they should be delineated as unitless.  Please provide interpretation of these ratios in terms of possible source influence.

Figures 4 and 5: Instead of figures, present a table of Spearman correlations of the VOCs.  (Spearman correlations should be used since the VOC data are likely not normally distrusted.)  One cannot interpret reported correlation (like reported in Line 168) by visually examining Figures 4 and 5.

Lines 155-164: As stated, wet samples were affected by rainfall or high humidity.  Analytical issues such as “FID flameouts” are mentioned regarding the lab analysis of the wet samples.  The authors need to provide information about how well resolved the VOC peaks were after the flameouts.

Lines 185-190: It is difficult to directly compare the well pad data to the Dallas Hinton site since benzene reported at Hinton were likely from canister samplers or auto-GCs, not passives as reported in this study.  Method differences could be a factor when comparing the study data to the Hinton site.  This is all the more reason why the Radiellos should have been compared with reference canister samplers.  If cans could not be deployed at the well pad site, then these comparisons should have been conducted at an air monitoring station near the university’s analytical lab.

Lines 191-195: My interpretation of Figure 3 is that there are no differences between summer and winter benzene values; there is a lot of scatter in the data.  The seasonal comparisons are mere hand-waving.  It would be better to show a summary table that compares minimum, median, and maximum VOC concentrations between a delineated “warm” season and a “cold” season.

Lines 200-204: As previously stated, winds were recorded from a site 30 km away from the monitoring site, a large distance.  An on-site, portable anemometer should have been used.  Including calm winds in the wind direction analysis is a potential error since this means winds were too weak to influence wind vane movement, and hence, direction.

Lines 228-230: Again, the Dallas Hinton data was likely from canisters (recorded once every 6-days), not from week-long Radiellos.  Method and time integral differences are a major factor when comparing the study data to Hinton.  This aside, the VOC data should also be compared to other Texas urban locations as well like the Houston Ship Channel sites (perhaps Clinton Drive); Ship Channel sites have petroleum refinery influence, in addition to traffic emissions.

Lines 218-230: The discussion of wind dispersion needs to include the limitation that the meteorology was measured from a site 30 km away and may not reflect actual conditions near the well pad.  (This is why it may be better to delete the wind direction analysis.)

Figures 7: Units were not provided for the wind speeds.

There is no Figure 8.

Figure 9 is a better presentation of benzene concentrations and temperature effects.  Therefore, I would suggest deleting Figure 3 as no consistent seasonal pattern can be inferred from that figure.

Reviewer 2 Report

The study shows a temporal behavior of some volatile organic compounds in South central Texas.

Lines 118-119. Authors don’t show results from experiments to evaluate analytical precision. Lines 124-125. Authors don’t show the results of the experiments on accuracy evaluation of the Radiello samplers. Authors don’t explain how they obtain the VOCs concentration. TD-GC-FID is an unspecific technique to identify unequivocally target compounds. It is well known that VOCs in the gas phase is a mixture extremely complex and FID is not suitable to completely identify target compounds. It is extremely necessary to use gas chromatography – mass spectrometry to confirm the presence of the target compounds. Authors don’t show the calibration curves. Authors don’t show any statistics on correlation analysis. Due to study just considered one sampling site, conclusions about health effect are weak. Authors should have been considered more sampling sites and less sampling days.

Reviewer 3 Report

This manuscript summarizes a pilot study testing a passive hydrocarbon sampler near an oil and gas site in the Eagle Ford Shale.  The paper is well written and provides useful information about the method, but has limited significance due to the very small sample size (2 active and 1 blank sampler at a single site).  It is not clear what the authors intend as the primary purpose of the paper:  a pilot test of a method, hypothesis-testing, or both?  As a methods paper, I think the manuscript is publishable with major revisions, but not as a hypothesis-testing paper since the sample size is much too small to draw any conclusions.  Below I have provided line edits that I suggest for improving the manuscript as a methods test paper:

Lines 27-33:  In your introductory paper, I would use the term "oil and natural gas" or "O&G" rather than "oil", since the wells are co-producing.  You could clarify in the second paragraph that some co-producing O&G sites flare gas because the wells became operational without sufficient natural gas gathering infrastructure.

Line 97:  Do you have evidence the flare occasionally was dysfunctional?  And can you provide more detail, such as was it unlit or smoking?

Lines 99-101:  What is the source of data for traffic? Does this include O&G heavy-duty diesel vehicles, which may have higher benzene emissions?

Section 2.2:  Provide more data on the saturation rates. At a minimum, provide preliminary data indicating that the expected saturation rate is much higher than measured.  Otherwise, the data could be erroneous due to over-saturation.

Lines 141-145:  Provide more information about the blank results.  What do you mean by absorbent-generated benzene? If you actually mean the sampler itself is a benzene source, then this seems like a major issue that must be resolved before the method is viable.

Table 1:  What is the time period of the maximum measured concentration?  A week?

Conclusion:  I suggest re-working the conclusion to focus on the methods rather than results.  You can summarize the results of the single sites, but do not generalize the results to other O&G sites without more data.  I would like to see more information about the cost and required expertise of the method.  For example, can you estimate how many samplers, time, and money would be required to collect sufficient data to estimate average benzene concentrations in the Eagle Ford?
